# Comparative Efficacy and Safety of Dupilumab and Benralizumab in Patients with Inadequately Controlled Asthma: A Systematic Review

**DOI:** 10.3390/ijms21030889

**Published:** 2020-01-30

**Authors:** Koichi Ando, Akihiko Tanaka, Hironori Sagara

**Affiliations:** Department of Medicine, Division of Respiratory Medicine and Allergology, Showa University School of Medicine, Shinagawa-ku, Tokyo 142-8666, Japan; tanakaa@med.showa-u.ac.jp (A.T.); sagarah@med.showa-u.ac.jp (H.S.)

**Keywords:** systematic review, asthma, dupilumab, benralizumab

## Abstract

No head-to-head trials have compared the efficacy and safety between the licensed dosage and administration dosage of dupilumab and benralizumab for inadequately controlled asthma. We conducted an indirect treatment comparison to estimate differences in the efficacy and safety between dupilumab and benralizumab for inadequately controlled asthma using the Bayesian approach. The primary efficacy endpoint was annual exacerbation rate (AER). A subgroup analysis by blood eosinophil count was also performed. The primary safety endpoint was the incidence of any adverse events (AAEs). The results demonstrate that there was no significant difference in the AER between dupilumab and benralizumab in overall patients and the subgroup with the blood eosinophil count of <150. However, the AER was significantly lower in the dupilumab group than in the benralizumab group in the subgroup with a blood eosinophil count of ≥150 but <300, and ≥300 with the rate ratio and 95% credible interval of 0.51 (0.29–0.92) and 0.58 (0.39–0.84), respectively. There was no significant difference in the AAEs between the dupilumab and benralizumab groups. This indirect treatment comparison indicates that dupilumab is superior to benralizumab in patients with inadequately controlled asthma having higher blood eosinophil counts. A direct comparison is required to provide definitive evidence. Systematic Review Registration: UMIN-CTR no. UMIN000036256.

## 1. Introduction

Bronchial asthma is estimated to affect 300 million patients worldwide, and thus, is a major global health and public health problem [1]. The Global Initiative for Asthma (GINA), a global asthma guideline, recommends a so-called “graded method” for increasing the number of inhaled steroids and β2 stimulants that are combined until adequate control is achieved to treat these asthma patients [2]. However, there are patients with severe asthma that cannot achieve asthma control (approximately 5% of all asthma patients) even with a combination of leukotriene receptor antagonist (LTRA), theophylline, and several oral asthma medications, such as oral corticosteroids, in addition to the maximum volume of inhaled corticosteroids (ICS)/long acting beta-2 stimulant (LABA) [3,4,5]. Several treatment options are already available and are widely used by patients with uncontrolled asthma [6,7,8]. These include biological agents such as anti-IgE antibody (omalizumab), anti-interleukin- (IL)-5 antibody (mepolizumab), and anti-IL-5 receptor α subunit -(IL-5Rα) antibody (benralizumab) [9], which are only effective for patients with specific characteristics, such as an atopic phenotype or eosinophilic phenotype; thus, these agents can only be used by a limited population [10,11]. That is, the patient population that can expect an improvement in lung function, a reduction in asthmatic attack frequency, an improvement in symptom score, etc., by these biological treatments is relatively limited. Therefore, there is a need for agents that can be highly effective for a broader patient population [12].

In this context, recently, an anti-IL-4 receptor α subunit (IL-4Rα) antibody (dupilumab) was approved and is available for patients with asthma when standard treatment cannot provide adequate asthma control. Dupilumab is profoundly involved in the amelioration of asthma by specifically binding to IL-4Rα shared by the IL-4 and IL-13 receptor complexes, inhibiting their signaling [13,14,15,16,17,18,19,20]. A recent phase III trial showed that dupilumab reduced the frequency of asthma exacerbation in not only asthma patients with high blood eosinophil count, but also in those with a relatively lower blood eosinophil count compared to patients administered conventional biologics [21]. These results suggest that dupilumab might be effective for a wide range of patients. However, although it is expected to be indicated in more asthma patients than conventional biological agents, some patients may be considered for treatment with multiple biological agents. For such patients, clinicians need to consider which biological agents to use (i.e., what biological agents are most likely to be effective).

Few direct head-to-head randomized controlled trials (RCTs) comparing the efficacy of dupilumab to existing biological agents have been performed. Thus, when clinicians consider the use of biologics for asthma patients for whom standard treatment does not provide adequate control, the data underlying drug selection are currently inadequate. A comparison of the efficacy and safety of multiple biological products to each other is therefore warranted [22], especially with dupilumab and existing antibody formulations such as mepolizumab and benralizumab. This is because data accumulated as a result of the comparison may provide clinicians with the guidance needed to select biological agents. Mepolizumab and benralizumab have been shown to be more effective for patients with higher blood eosinophil count (blood eosinophil count ≥300) [23,24,25,26]. Thus, the results of a study comparing dupilumab to biological agents such as mepolizumab and benralizumab via a blood eosinophil count subgroup, are expected to more clearly reveal the factors that are important when making formulation choices.

We specifically focused on comparing the efficacy of dupilumab and benralizumab, which are receptor subunit antibodies, on inadequately controlled asthma [27,28]. From a clinical application viewpoint, it was desirable that these comparisons be performed at the licensed dosage and administration. Furthermore, analysis results comparing these drugs by blood eosinophil count were considered useful information for clinicians to select drugs based on blood eosinophil count. With this point of view, we conducted a Bayesian network meta-analysis and compared the efficacy and safety of licensed doses of dupilumab and benralizumab administered to uncontrolled, persistent asthma cases via a blood eosinophil count subgroup. Although head-to-head RCTs may be the preferred method to compare efficacy and safety between drugs, they are generally costly, time consuming, and tedious to perform. In contrast, indirect treatment comparisons (ITCs) can be performed in the absence of a head-to-head RCT and may deliver much needed information to clinicians more quickly than RCTs [29]. Using a Bayesian network meta-analysis, it is possible to perform indirect comparison via a common comparison drug (usually a placebo group) [29]. If there is no direct comparison, the results of the analysis are expected to provide useful evidence for clinicians to select biological agents. The aim of the present study was to indirectly compare the efficacy and safety of dupilumab and benralizumab at the licensed dosage and administration by blood eosinophil counts using the statistical method of the Bayesian network meta-analysis.

## 2. Results

### 2.1. Systematic Review

By performing a systematic literature review, 557 studies were identified (348 from PubMed, 146 from Embase, 31 from CENTRAL, and 32 from SCOPUS), of which 436 articles were retained after duplicates were removed. Among these, three studies were identified based on the patients, interventions, comparison, outcomes, and study design (PICOS) criteria; these included one dupilumab study (LIVERTY ASTHMA QUEST) [21] and two benralizumab studies (CALIMA and SIROCCO [23,24]). These three studies were considered eligible and were included in the analysis. The study selection process is shown in Figure 1.

The main characteristics of the included studies are shown in Appendix A. In these RCTs, there were multiple treatment arms set for different dosages and administrations, but only the arms of licensed dosage and administration and matched placebo arms (total number of patients = 2640) were included in the present analysis. Although data of the included studies were available for an ITC analysis of one pre-defined primary efficacy endpoint (AER), there were insufficient data for an ITC analysis of two pre-defined secondary efficacy endpoints (change in forced expiratory volume at 1.0 s (FEV1.0) and AQLQ score). This is because the data reported in the included studies for the licensed dose of dupilumab were insufficient to analyze the patient group with the blood eosinophil count of ≥150 but <300, and <150. The ITC analysis of these secondary efficacy endpoints was performed only in the overall population and in the subpopulation with a blood eosinophil count of ≥300. This did not pose a problem for the comparison of efficacy between dupilumab and benralizumab in patients with a higher blood eosinophil count, which we were most interested in. As a result of this assessment, the model convergence condition using the BCR diagnostic method, the preferred model convergence, was confirmed in the analysis. A network map of this ITC is presented in Figure 2.

### 2.2. Primary Efficacy Endpoint

#### AER

Dupilumab and benralizumab reduced the AER compared with that of the placebo with the respective rate ratio (RR) and 95% credible interval (CrI) of 0.54 (0.43–0.67) and 0.65 (0.55–0.77) in the overall population and 0.32 (0.24–0.45) and 0.57 (0.46–0.70) in the subgroup with the blood eosinophil count of ≥300 (Figure 3A,B). In the subgroup with blood eosinophil ≥150 and <300, dupilumab reduced AER compared to placebo with RR and 95% CrI of 0.40 (0.26 to 0.61), whereas benralizumab did not show a significant AER difference compared to placebo (RR and 95% CrI of 0.77 (0.52 to 1.15); Figure 3C). In the sub-group with a blood eosinophil count of <150, neither dupilumab nor benralizumab showed a significant difference in AER compared to the placebo with RR and 95% CrI of 1.15 (0.75 to 1.72) and 0.73 (0.48 to 1.10), respectively (Figure 3D). The comparison between the drugs showed that AER was significantly better in the dupilumab group than the benralizumab group for the subgroup with a blood eosinophil count of ≥300 and a blood eosinophil count of ≥150 but <300 with RR and 95% CrI of 0.58 (0.39 to 0.84) and 0.51 (0.29 to 0.92), respectively (Figure 3B,C). We found no significant difference in the AER between both drugs in the overall population and in the subgroup with the blood eosinophil count of <150 with the RR and 95% CrI of 0.83 (0.62–1.09) and 1.57 (0.73–2.82), respectively (Figure 3A,D).

### 2.3. Secondary Efficacy Endpoint

#### Changes in FEV1.0 and AQLQ Score from Baseline

The changes in FEV1.0 from the baseline for the dupilumab and benralizumab groups were significantly better than those for the placebo with the respective mean difference (MD) and 95% CrI of 0.130 (0.068–0.194) and 0.099 (0.051–0.146) in the overall population, and 0.251 (0.155–0.347) and 0.146 (0.088–0.204) in the subgroup with the blood eosinophil count of ≥300 (Figure 4A,B). The comparison of the two drugs showed no significant difference in the change in FEV1.0 from the baseline in the overall population and the subgroup with the blood eosinophil count of ≥300 (Figure 4A,B).

The changes in AQLQ score from baseline in the dupilumab group and benralizumab group were significantly better than the placebo with respective MD and 95% CrI of 0.261(0.111 to 0.408) and 0.220 (0.106 to 0.333) in the overall population, and 0.342 (0.120 to 0.565) and 0.300 (0.161 to 0.439) in the subgroup with a blood eosinophil count of ≥300 (Figure 5A,B). The comparison between the two drugs showed no significant difference in the AQLQ score from the baseline in the overall population and the subgroup with the blood eosinophil count of ≥300 with the MD and 95% CrI of 0.041 (−0.145 to 0.227) and 0.042 (−0.220 to 0.304), respectively (Figure 5A,B).

### 2.4. Primary and Secondary Safety Endpoint

#### Incidence of AAE and SAE

There were no significant differences in the incidence of AAEs between dupilumab or benralizumab and placebo, with the odds ratio (OR) and 95% CrI of 0.830 (0.591–1.165) and 0.811 (0.619–1.061), respectively, and between dupilumab and benralizumab with the OR and 95% CrI of 1.023 (0.688–1.526) (Figure 6A), and there were no significant differences in the incidence of any SAEs between dupilumab or benralizumab and placebo, with OR and 95% CI of 1.039 (0.657 to 1.639) and 0.787 (0.550 to 1.129), respectively, and between dupilumab and benralizumab, with 1.319 (0.768–2.265) (Figure 6B).

### 2.5. Bias Assessment

Evaluating the risk of bias using the Cochrane risk of bias tool revealed a low risk of bias for all studies included in this analysis. The risk of bias graph and risk of bias summary are presented in Figure 7A,B.

## 3. Discussion

The present ITC compared the efficacy and safety of dupilumab and benralizumab in patients with inadequately controlled asthma. The studies included in this ITC were obtained via a systematic literature review conducted in accordance with generally accepted methodologies (i.e., Preferred Reporting Items for Systematic reviews [PRISMA] guidelines) [30] and further adapted the PICOS framework eligibility criteria. The licensed dosage and administration of dupilumab and benralizumab in patients with uncontrolled asthma were interpreted using the statistical method of Bayesian network meta-analysis based on the results of the included phase III studies. Currently, there are several biologics available for patients with uncontrolled asthma who are often eligible for multiple biological treatments. It is important and meaningful to compare the efficacy and safety of dupilumab and benralizumab, which can help clinicians in drug prescription. A recent review of severe asthma emphasized the importance of comparing the effects of biological agents [22].

In the overall population, AER, which is the primary endpoint, did not differ significantly between dupilumab and benralizumab. A subgroup analysis by eosinophil count also showed no significant difference in AER between dupilumab and benralizumab in the group of <150; however, in the subgroup with blood eosinophil count ≥150 and <300, and ≥300, AER was significantly better in the dupilumab group compared to the benralizumab group. Changes in FEV1.0 and AQLQ score from baseline were analyzed in the overall population and in the subgroup with blood eosinophil count of ≥300. The results reveal that the changes from baseline (i.e., FEV1.0 and AQLQ score) were significantly better than placebo, with no significant difference between the two drugs. Regarding safety profiles, there was no significant difference between the two drugs for the incidence of AAEs, any SAE, and death caused by AAEs.

In a phase III trial included in this study, dupilumab and benralizumab significantly improved AER compared to placebo in the patient group with blood eosinophil count of ≥300. In the group with blood eosinophils of ≥150 but <300, dupilumab had a significantly better AER compared to placebo, but there was no significant difference in AER compared to placebo for benralizumab in this subgroup [21,23,24]. As expected, in the group with blood eosinophils ≥150 and <300, dupilumab displayed a better efficacy profile for AER than benralizumab. Although there is very little evidence on the comparison of dupilumab and benralizumab in groups with >300 blood eosinophils, the results of our indirect comparison, for the first time, indicate that dupilumab significantly improve AER compared to benralizumab in the group with blood eosinophil count of ≥300.

Similarly, in a phase III trial of uncontrolled persistent asthma which we included in this study [21,23,24], dupilumab and benralizumab statistically significantly improved FEV1.0 and AQLQ scores compared to placebo, respectively, particularly in the group with blood eosinophil count of >300. In our indirect comparison, data of FEV1.0 and AQLQ scores from the included studies were available only in the analysis of the overall population and subgroups with blood eosinophil count of ≥300. Although we did not find any significant difference in FEV1.0 and AQLQ scores between these drugs, this indirect comparison revealed that in terms of change in FEV1.0, dupilumab had a better tendency than benralizumab in subgroups with blood eosinophil count of ≥300. Although the safety profiles of both treatments are similar, eosinophilia was reported as a minor adverse event in the Dupilumab Phase III trial, specifically in the dupilumab group. These results indicate that additional clinical studies are needed to assess the safety of dupilumab.

In this analysis, dupilumab showed a better efficacy profile than benralizumab, especially in the high blood eosinophil group. This result may be explained by molecular biology. The IL-5 receptor is a protein complex belonging to the type I cytokine receptor family and is located at the cell membrane [31]. Various signaling pathways including Ras-MAP kinase, JAK-STAT, PI3-kinase, and Src family kinases are activated by the binding of the IL-5 ligand to the extracellular domain of the IL-5 receptor [31,32]. These signaling pathways play a key role in infiltration into the airway tissue and are implicated in the pathogenesis of bronchial asthma [32]. Benralizumab is a fucose-deficient humanized immunoglobulin G subclass 1, κ isotype (IgG1κ) monoclonal antibody that specifically shows a high binding affinity for human IL-5Rα [33]. Due to fucose deficiency in the Fc domain, benralizumab exhibits a high affinity (dissociation constant: 45.5 nM) for FcγRIIIa. Benralizumab induces antibody-dependent cytotoxicity, which triggers apoptosis of eosinophils and basophils expressing IL-5Rα [33,34]. However, type 2 inflammation, which is closely related to the intractability and severity of bronchial asthma, is known to involve not only IL-5 and its downstream signals but also various inflammatory cytokines. IL-5 plays only a limited role in mediating type 2 inflammation [13,15]. In contrast, IL-4 and IL-13 are chiefly produced by CD4+ Th2 and type 2 innate lymphoid cells. They play a key role in type 2 inflammation and are predominantly responsible for many changes, including inflammatory and structural changes that characterize the pathogenesis of asthma [35,36,37]. In fact, IL-4 and IL-13 promote the switching of Ig class from IgM antibodies to IgE in the B lymphocytes, induce retractility of airway smooth muscle via synthesis of eotaxin, and recruit eosinophils to the airways [38,39,40]. IL-13 also enhances airway remodeling in asthma by inducing the transformation of bronchial fibroblasts to myofibroblasts; in addition, it promotes hyperplasia of goblet cells, stimulates proliferation of airway smooth muscle cells, and induces collagen deposition (Figure 8).

A heterodimeric receptor complex consisting of IL-4Rα and IL-13 receptor α1 subunit (IL-13Rα1) is activated by IL-4 and IL-13. These heterodimeric receptor complexes are not only frequently expressed in eosinophils, but also in basophils, lymphocytes, monocytes/macrophages, dendritic cells, fibroblasts, bronchial epithelial cells, endothelial cells; these play an important biological role [41,42,43,44]. IL-5 is a cytokine with high specificity for eosinophils while IL-4 and IL-13 are highly diverse cytokines that not only affect inflammatory cells, but also constitutive cells. In our indirect comparison, dupilumab resulted in a relatively better improvement, although not significant, in respiratory function than benralizumab in subgroup with higher blood eosinophil counts. This may reflect the effect of dupilumab on bronchial smooth muscle. IL-4 and IL-13 play a central role in the pathophysiology of asthma and could be very promising and appropriate targets for specific and effective new therapies [45,46]. The molecular biological mechanisms may explain the superiority of dupilumab to benralizumab. As this analysis is an indirect comparison, it is rather difficult to draw definitive conclusions from the results attained; however, they may suggest the superiority of dupilumab to benralizumab in patients with inadequately controlled asthma. As asthma is a rather diverse disease, patients with high blood eosinophil counts may not display a better efficacy with dupilumab than with benralizumab. In addition, in a specific patient population, benralizumab may be more effective than dupilumab. Therefore, future research that seeks to find the characteristics of patients that display more effective results with benralizumab than dupilumab might be desired.

Our present ITC had several limitations. First, there were inconsistencies between studies. Moderate to severe bronchial asthma patients diagnosed by the GINA guidelines were included in the Dupilumab Phase III trial but only patients with severe illness were included in the benralizumab group. That is, the inclusion criteria with respect to disease severity differed between the two drugs. In particular, the benralizumab group had at least two exacerbations per year, while the dupilumab group had at least one exacerbation per year. Biologics have been reported to be particularly effective in patients with more severe disease and in those with more frequent exacerbations. If the severity or frequency of exacerbations is standardized among the studies, the difference in efficacy between the two drug groups is expected to be more pronounced. Thus, heterogeneity among studies was not considered to affect the final conclusion of our analysis, which suggested that dupilumab is superior to benralizumab. Second, an analysis was performed for multiple endpoints, but the asthma control questionnaire (ACQ) score and the reduction effect of hospitalization could not be included as endpoints. The AER, which is the endpoint included in this analysis, is considered to be the most important and a highly relevant endpoint, not only for the patient’s medical condition, but also from the aspect of healthcare economics.

In summary, this study compared the efficacy and safety of dupilumab and benralizumab in patients with poorly controlled persistent asthma. In the eosinophil high level group, dupilumab was found to display a better efficacy profile than benralizumab and was generally well tolerated. Considering that this analysis is an indirect comparison, a further verification, such as an RCT by direct comparison, is required to confirm the results reported herein.

## 4. Materials and Methods

### 4.1. Systematic Review

To identify published RCTs of dupilumab and benralizumab from 1946 to the present, we conducted a comprehensive literature search of PubMed (MEDLINE), Embase, Cochrane Central Register of Controlled Trials (CENTRAL), and SCOPUS using search terms such as “asthma”, “dupilumab”, and “benralizumab”. The search strategy for PubMed is described in Appendix A. The same search strategy was used to search Embase, CENTRAL, and SCOPUS. This systematic review of the literature is mainly intended to identify all publicly available RCTs to support the comparison of efficacy and safety between dupilumab and benralizumab in patients with inadequately controlled asthma. To identify all relevant eligible studies and minimize publication bias, a review of the references presented in the articles and a manual search of relevant articles were performed. This systematic review process was conducted in accordance with PRISMA guidelines [30]. The inclusion and exclusion criteria for predefined PICOS were adapted for the studies retrieved in this systematic review to address the clinical or methodological heterogeneity between studies and to ensure quality of the indirect comparison analysis.

### 4.2. Quality Evaluation

The risk of bias tool recommended by the Cochrane Collaboration [47] was used to assess the qualities of RCTs included in the present analysis. Several parameters: (1) random sequence generation; (2) allocation concealment; (3) blinding of participants and personnel; (4) blinding of outcome assessment; (5) incomplete outcome data; (6) selective reporting; and/or (7) other bias, were assessed as high, unclear, or low.

### 4.3. Inclusion and Exclusion Criteria (Pre-Defined PICOS)

#### Patients

To be included in this analysis, the patient group had to meet the following criteria: adolescent or adult patients with asthma who met the GINA guidelines diagnostic criteria of 12 years of age or older [2]; patients with moderate-to-severe persistent asthma who received 200 μg/day fluticasone or an equivalent or more ICS with at least one clinically significant episode (require administration of systemic steroids or consultation at an emergency outpatient center or admission); FEV1.0 before bronchodilator administration of less than 80% (an adolescent with less than 90% was acceptable); FEV1.0 reversibility after administration of short-acting beta-2 agonist of ≥12%, or ≥200 mL; and the ACQ score of ≥1.5 before inclusion. The exclusion criteria were patients under 12 years of age, current smokers, or COPD patients.

### 4.4. Interventions/Comparisons

Treatments eligible for this analysis (ITC) were dupilumab 300 mg every 2 weeks (first administration with 600 mg as loading dose) and benralizumab 30 mg every 8 weeks (every 4 weeks for the first three administrations), which are licensed dosages and administrations; included studies should have had either of these as a treatment group. The common comparative group in the dupilumab studies and benralizumab studies was assumed to be placebo.

### 4.5. Outcomes

The primary efficacy endpoint was the frequency of clinically significant exacerbations. Clinically significant exacerbation was defined as exacerbation requiring systemic corticosteroid treatment, hospital admission, or emergency clinic visits. The result is expressed as AER. Secondary efficacy endpoints were changes in FEV1.0 and asthma quality of life questionnaire (AQLQ) score from baseline. In addition to an analysis of the overall population, a subgroup analysis by blood eosinophil count was performed in each group with blood eosinophils of ≥300, ≥150 but <300, and <150 for these primary or secondary efficacy outcomes. The primary safety endpoint was incidence of any adverse event (AAE), and the secondary safety endpoint was incidence of SAE. These pre-defined endpoints were analyzed only if the data reported in the included studies were available.

### 4.6. Study Design

Studies eligible for inclusion in this analysis were defined as phase III studies in a double-blind, parallel group RCT. One or more preset efficacy or safety endpoints should have also been available for the analysis.

### 4.7. Statistical Analysis Method of Indirect Comparison

ITCs of dupilumab (300 mg administered subcutaneously, 600 mg administered subcutaneously as loading dose, every 2 weeks) and benralizumab (30 mg subcutaneously every 8 weeks but every 4 weeks for the first three administrations) for each predefined endpoint via the common comparator of placebo was performed using the Bayesian network meta-analysis statistical method in accordance with the established methodology outlined by the National Institute for Health and Care [48,49,50]. This method is supported not only by the International Society for Pharmacoeconomics and Outcome Research (ISPOR) guidelines for indirect comparison and network meta-analysis, but the National Institute for Health and Clinical Excellence and Haute Autorité de Santé [51,52]. In the absence of direct comparison RCTs, the methodology of indirect comparison is useful for comparing treatment regimens [29]. In this analysis, we performed an indirect comparison by using statistical methods of Bayesian network meta-analysis with the random effect model. This statistical method is considered to be a well established statistical method of ITC [53,54], and in a broad range of fields, many reports of ITC using this statistical method have been described previously [55,56,57,58,59,60,61]. An analysis was performed not only in the overall population, but in the limited population with blood eosinophils of ≥300, ≥150 but <300, and <150 as subgroup analysis. In this Bayesian analysis, the posterior distribution of treatment effects was estimated by Gibbs sampling using the Markov chain Monte Carlo method. The treatment effect is expressed as the RR, MD, and OR with 95% CrIs. If the number of included clinical trials is small and the number of subjects is high, the National Institute for Health and Clinical Excellence recommends that a non-informative prior distribution be used for Bayesian analysis [62]. The present analysis was intended to be based on Phase III trials and was assumed to involve many patients. However, the number of included studies was limited and the non-informative prior distribution was assumed for all treatment effects. We used Gould’s method in the present ITC [63]. Iteration was performed 50,000 times with the first 10,000 taken as burn-in samples. We derived 95% CrI from 2.5 and 97.5% of the posterior distribution. Results were interpreted as not significant if the 95% CrI exceeded the ineffective line (i.e., if the RR or OR was 1 and MD was 0). The Brooks-Gelman-Rubin (BGR) diagnostic method was used to assess model convergence [64,65]. The analysis was performed using OpenBUGS 1.4.0 (MRC Biostatistics Unit, Cambridge Public Health Research Institute, Cambridge, UK). STATA (ver14.College Station, TX) was used to create graphics for the presentation of the results.

### 4.8. Ethical Aspects

Institutional review board approval and patient consent were not required and were thus waived due to the nature of the review performed in this study.

## 5. Conclusions

In this study, we compared the efficacy and safety of dupilumab and benralizumab in patients with inadequately controlled asthma. Dupilumab revealed a better efficacy profile than benralizumab in the group with a high eosinophil count, and it was generally well tolerated. Considering that this analysis is an indirect comparison, a further analysis, such as an RCT by direct comparison, is required to confirm the results reported herein.

## Figures and Tables

**Figure 1 ijms-21-00889-f001:**
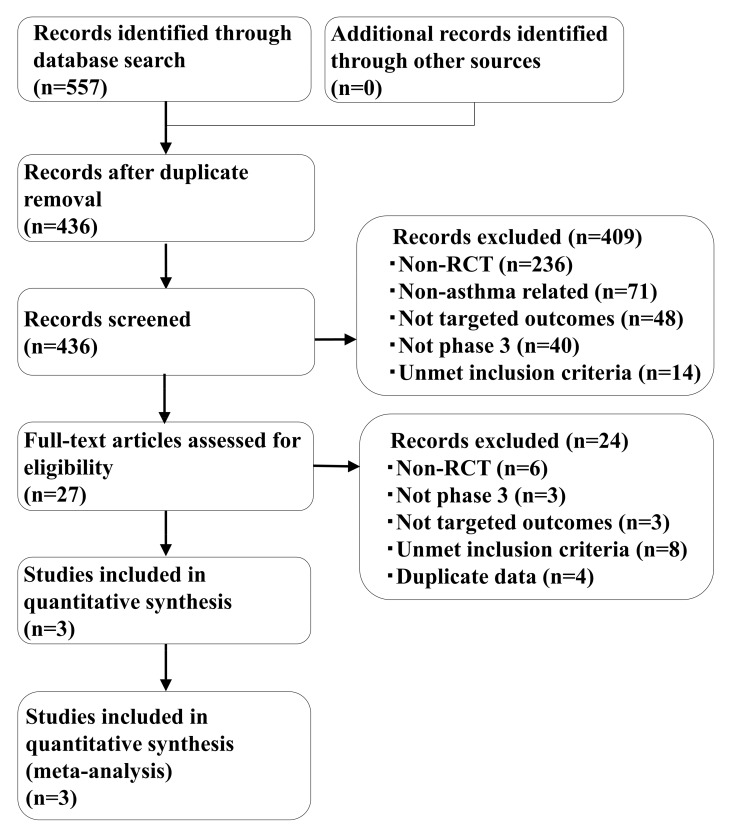
Preferred reporting items for systematic reviews and meta-analyses flow diagram of the study selection.

**Figure 2 ijms-21-00889-f002:**
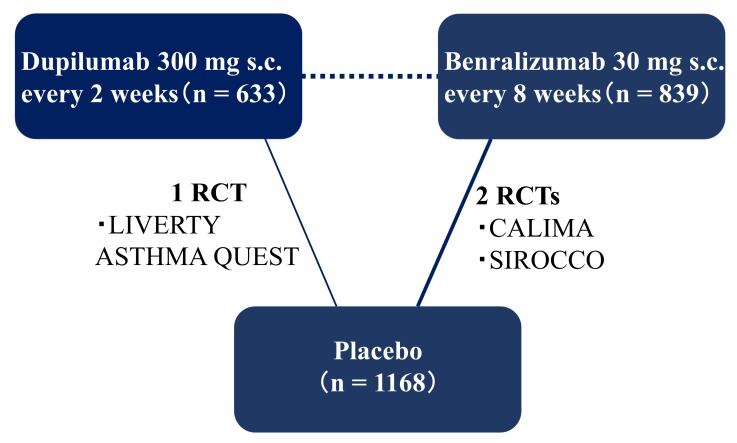
Network of the indirect treatment comparison performed in this study. The randomized controlled trials included in this analysis are shown as solid lines with their thickness representing the number of included studies. s.c., subcutaneous; RCT, randomized controlled trial.

**Figure 3 ijms-21-00889-f003:**
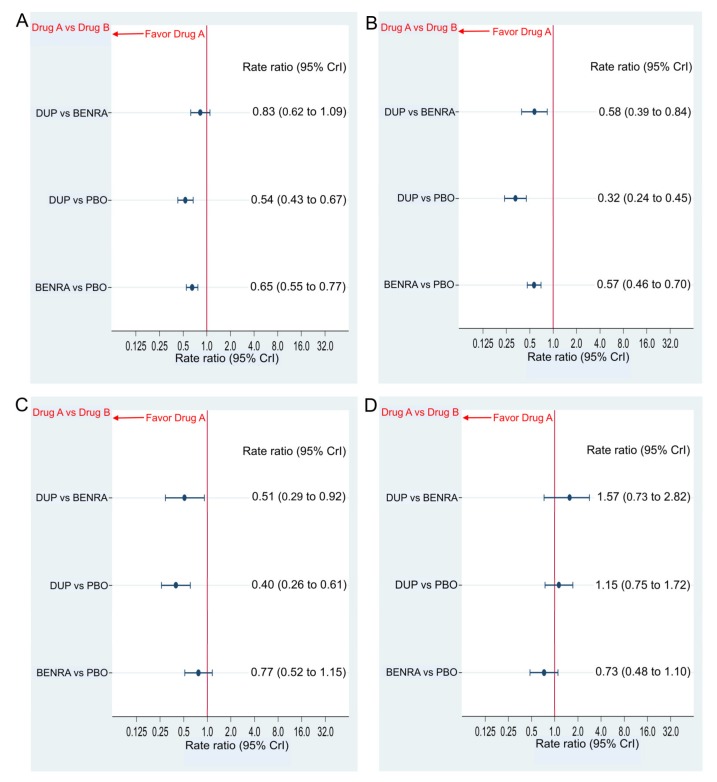
Comparative efficacy in annualized exacerbation rate (AER) of dupilumab 300 mg (loading dose, 600 mg) when administered subcutaneously every 2 weeks and benralizumab 30 mg administered subcutaneously every 8 weeks (first three doses administered every 4 weeks). Data are expressed as rate ratio and 95% credible intervals (CrIs). (**A**) Comparison in the overall population, (**B**) comparison in the patient subgroup with the blood eosinophil count of ≥300, (**C**) comparison in the patient group with the blood eosinophil count of ≥150 but <300, (**D**) comparison in the patient group with the blood eosinophil count of <150; DUP, dupilumab; BENRA, benralizumab; PBO, placebo.

**Figure 4 ijms-21-00889-f004:**
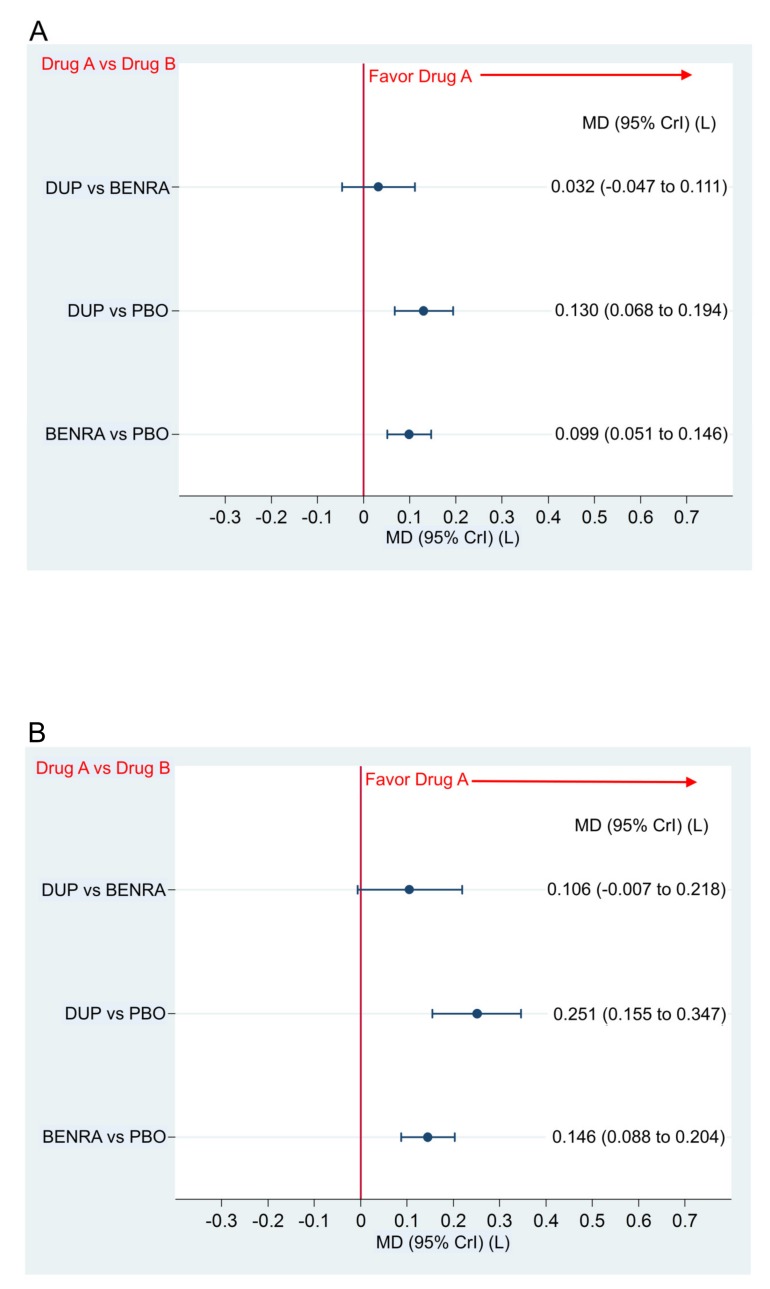
Comparative efficacy in the change in FEV_1.0_ of dupilumab 300 mg (loading dose, 600 mg) when administered subcutaneously every 2 weeks and benralizumab 30 mg administered every 8 weeks (first three doses administered every 4 weeks). Data are expressed as mean difference (MD) (L) and 95% credible interval (CrI). (**A**) Comparison in the overall population, (**B**) comparison in the patient subgroup with the blood eosinophil count of ≥300; DUP, dupilumab; BENRA, benralizumab; PBO, placebo.

**Figure 5 ijms-21-00889-f005:**
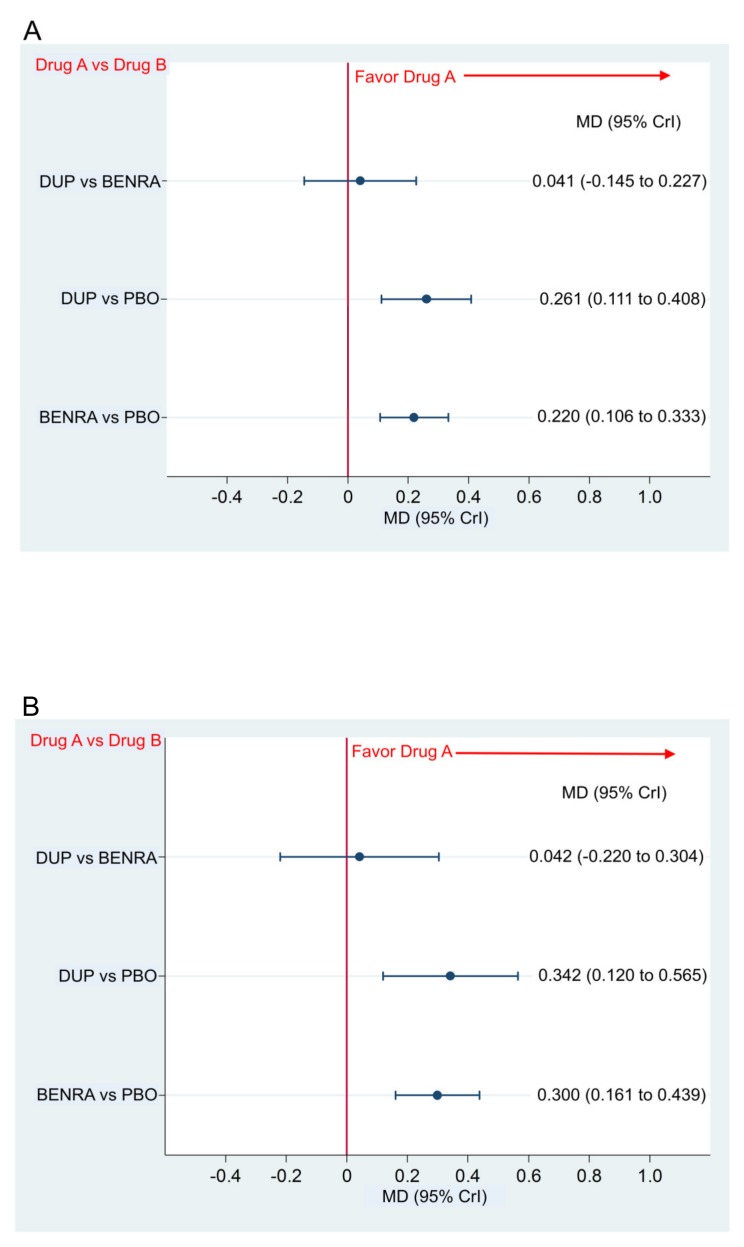
Comparative efficacy in the change in AQLQ score of dupilumab 300 mg (loading dose, 600 mg) when administered subcutaneously every 2 weeks and benralizumab 30 mg administered every 8 weeks (first three doses administered every 4 weeks). Data are expressed as mean difference (MD) and 95% credible interval (CrI). (**A**) Comparison in the overall population, (**B**) comparison in the patient subgroup with the blood eosinophil count of ≥300; DUP, dupilumab; BENRA, benralizumab; PBO, placebo.

**Figure 6 ijms-21-00889-f006:**
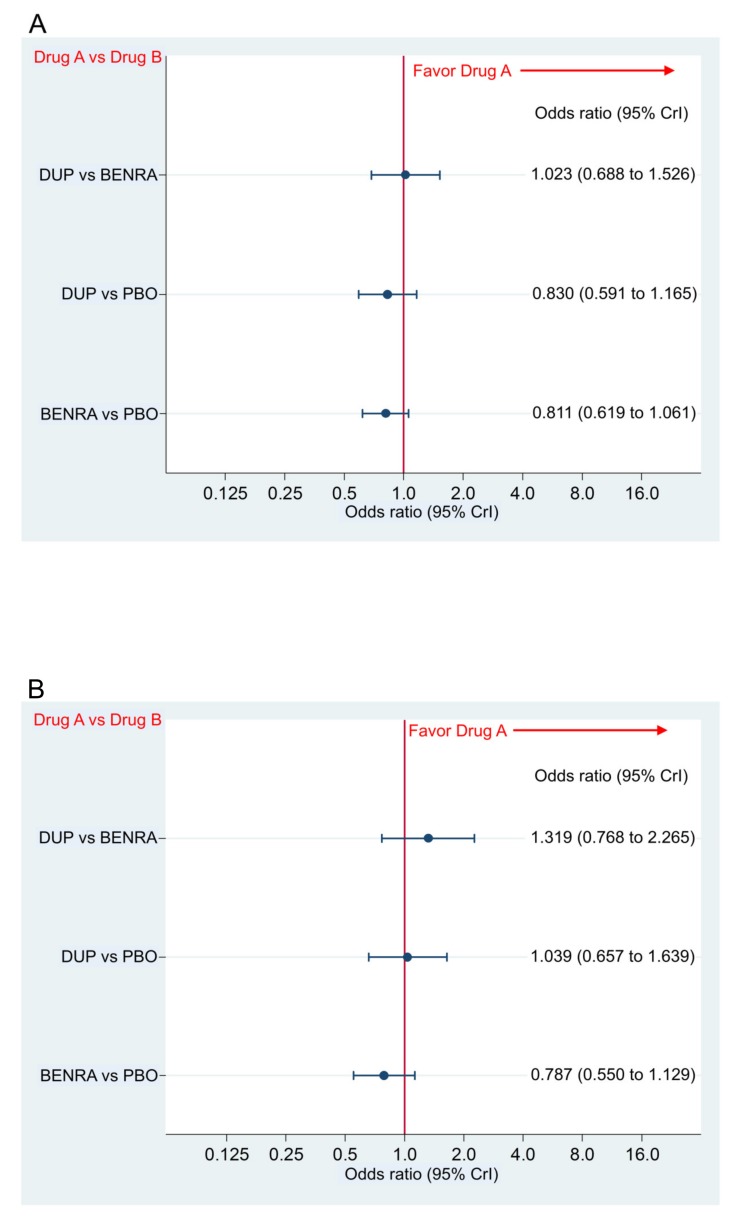
Comparative safety in the incidence of any adverse events (**A**), and any severe adverse events (**B**) of dupilumab 300 mg (loading dose, 600 mg) when administered subcutaneously every 2 weeks and benralizumab 30 mg administered every 8 weeks (first three doses administered every 4 weeks), in the overall population. Data are expressed as odds ratio and 95% credible interval (CrI); DUP, dupilumab; BENRA, benralizumab; PBO, placebo.

**Figure 7 ijms-21-00889-f007:**
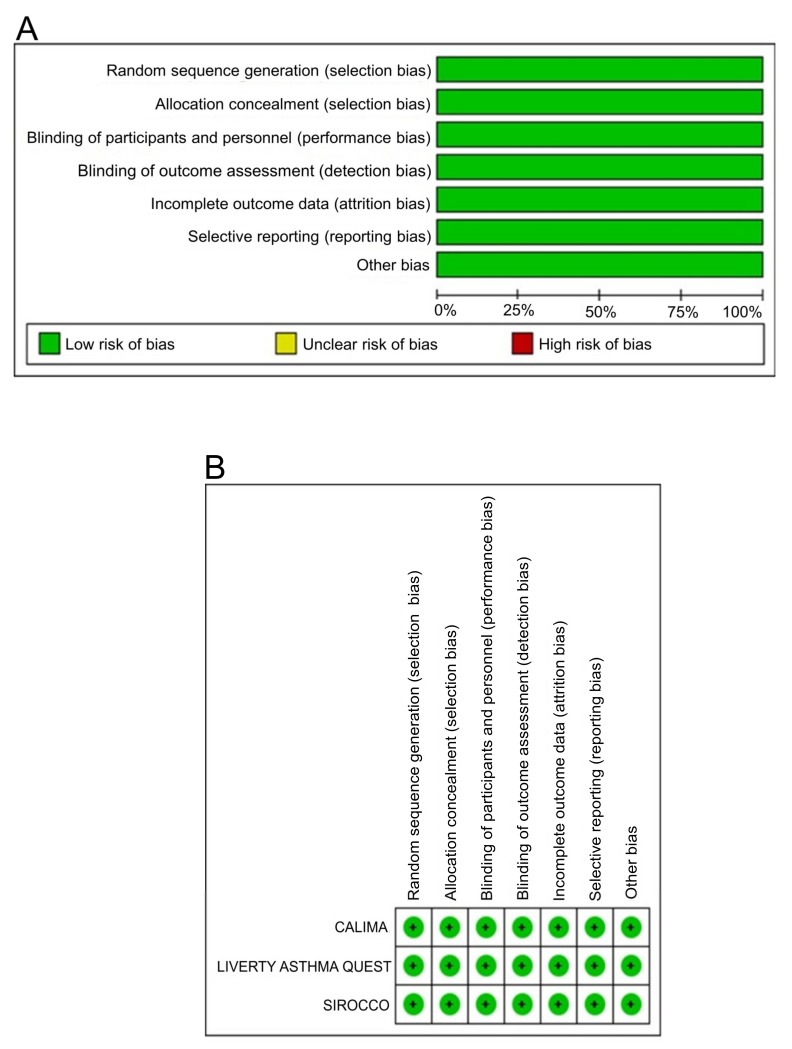
Risk of bias graph (**A**): review of authors’ judgements regarding each risk of bias item presented as percentages across all included studies. The risk of bias summary (**B**): review of authors’ judgements regarding each risk of bias item for each included study.

**Figure 8 ijms-21-00889-f008:**
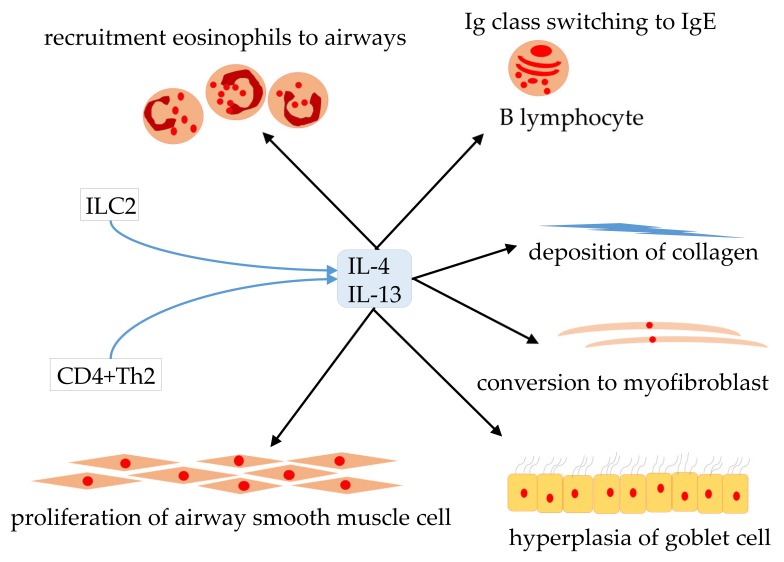
IL-4 and IL-13 are chiefly produced by CD4 + Th2 and type 2 innate lymphocytes. They play a vital role in type 2 inflammation, which is deeply involved in the typical inflammation as well as structural changes that characterize the pathogenesis of asthma.

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
