# Peer review of "Comparative Efficacy and Safety of Dupilumab and Benralizumab in Patients with Inadequately Controlled Asthma: A Systematic Review"

_ijms, 2020, doi:10.3390/ijms21030889_

Round 1
Reviewer 1 Report
Overall the paper makes an important contribution to understanding the effectiveness of treatment for uncontrolled asthma. The methodology used is robust and the conclusions acknowledge the limitations of not doing a direct phase 3 trial. Overall I only have minor corrections to the paper and would suggest a less definitive conclusion in the abstract based on the limitations. The references to figures in the results also appears to be incorrect.
Abstract
Line 10: should read “between the licensed dosage” Can the conclusion “this indirect treatment comparison indicates that dupilumab is superior to benralizumab in patients with higher blood eosinophil count” also note the patient group (individuals with uncontrolled asthma) and note the need for direct comparisons to confirm the effect.
Introduction
Add a comma after “agents” and before “such” at line 37 and at line 39 between “characteristics” and “such”. Line 39 add “an” before “atopic phenotype” Line 41 add “an” between “can” and “improvement”Results
Please note the final number of studies included in the analysis in line 89-90. Line 103: it appears there is a duplicate of “forced expiratory volume at 1.0 second”. Line 121: the reference to “Figures 1 and 2 seems incorrect (throughout this paragraph). Please review. Should these be 3a and 3b?No further comments
Author Response
Response to Review report 1
Thank you for giving me the opportunity to submit a revised draft of my manuscript titled “Comparative efficacy and safety of dupilumab and benralizumab in patients with inadequately controlled asthma: a systematic review” to International journal of molecular sciences. We appreciate the time and effort that you have dedicated to providing your valuable feedback on my manuscript. We are grateful for your insightful comments on my paper. We have been able to incorporate changes to reflect most of the suggestions provided by the reviewers. We have highlighted (yellow) the changes within the manuscript.
Here is a point-by-point response to the reviewers’ comments and concerns.
Comment 1:
Overall the paper makes an important contribution to understanding the effectiveness of treatment for uncontrolled asthma. The methodology used is robust and the conclusions acknowledge the limitations of not doing a direct phase 3 trial. Overall I only have minor corrections to the paper and would suggest a less definitive conclusion in the abstract based on the limitations. The references to figures in the results also appears to be incorrect.
Response: We agree with this and have incorporated your suggestion. Line 23-24: In conclusion part of abstract section, we noted the need for direct comparisons to confirm the results of this study because our study was indirect treatment comparison. The references to figures in the results were revised (Line 122).
Comment 2:
Abstract
Line 10: should read “between the licensed dosage” Can the conclusion “this indirect treatment comparison indicates that dupilumab is superior to benralizumab in patients with higher blood eosinophil count” also note the patient group (individuals with uncontrolled asthma) and note the need for direct comparisons to confirm the effect.
Response: We agree with this and have incorporated your suggestion. Thank you very much for the very important suggestions. As suggested, we have corrected the relevant points. In the conclusion part, we have mentioned about the patient group and have stated that a direct comparison is necessary to confirm the results obtained in this indirect comparison. Line22 and later have been revised as follows.
” This indirect treatment comparison indicates that dupilumab is superior to benralizumab in patients with inadequately controlled asthma having higher blood eosinophil count. Direct comparison is required to provide definitive evidence”
Comment 3:
Introduction
Add a comma after “agents” and before “such” at line 37 and at line 39 between “characteristics” and “such”. Line 39 add “an” before “atopic phenotype” Line 41 add “an” between “can” and “improvement”
Response: Agree. Thank you for pointing this out. All the suggested changes have been incorporated in the revised manuscript as follows:
Line 37: A comma has been added after “agents” and before “such”.
Line 39: A comma has been added between “characteristics” and “such”.
Line 39: We have added “an” before “atopic phenotype”.
Line 41: We have added “an” between “expect” and “improvement”.
Comment 4:
Results
Please note the final number of studies included in the analysis in line 89-90. Line 103: it appears there is a duplicate of “forced expiratory volume at 1.0 second”. Line 121: the reference to “Figures 1 and 2 seems incorrect (throughout this paragraph). Please review. Should these be 3a and 3b?
Response: Agree. Thank you for pointing this out. All the suggested changes have been incorporated in the revised manuscript as follows:
Line 90-93: We have noted the final number of studies included in the analysis.
Line 103: The duplicate sentence “force expiratory volume at 1s” has been removed.
Line 122: “(Figures 1 and 2)” have been changed to “(Figure 3A, 3B)”
Line 130: “(Figure 2B, C)” has been changed to “(Figure 3B, 3C)”
Line 133: “(Figure 3)” has been changed to “(Figure 3A, 3D)”
In addition to the above comments, all spelling and grammatical errors pointed out by the reviewers have been corrected.
We look forward to hearing from you in due time regarding our submission and to respond to any further questions and comments you may have.
Sincerely,
Koichi Ando
Department of Medicine, Division of Respiratory Medicine and Allergology, Showa University School of Medicine
1-5-8 Hatanodai, Shinagawa-ku, Tokyo, 142-8555, Japan
Tel: +81-3784-8532
Fax: +81-3-3784-8742
Email: koichi-a@med.showa-u.ac.jp
Reviewer 2 Report
This is a systemic literature review, it focused on comparing the efficacy and safety between dupilumab and benralizumab for unadequately controlled asthma. Initial 557 studies were enrolled and through selection process, only one dupilumab study and two benralizumab studies were identified after patients, interventions, comparison, outcome, and study designs (PICOS) were adopted. Using the Bayesian network meta-analysis statistical method with non-informative prior distribution, the results demonstrated that there was no significant difference in the annual exacerbation rate (AER) between dupilumab and benralizumab in overall patients and the subgroup with the blood eosinophil count of <150. However, the AER was significantly lower in the dupilumab group than in the benralizumab group in the subgroup with the blood eosinophil count of ≥150 but <300, and ≥300 with the rate ratio and 95% credible interval of 0.51 (0.29–0.92) and 0.58 (0.39–0.84), respectively. This indirect treatment comparison indicates that dupilumab is superior to benralizumab in patients with higher blood eosinophil count. This is the first paper to propose non-head-to-head comparing the efficacy and safety of dupilumab and benralizumab. Although the results were well described, however there are still some issues that must be clarified.
Major:
Although the authors mentioned that there were quite high-quality papers in this analysis, they also think that there was bias in the analysis of this study, but because their inclusion criteria were different in the severity degree, especially the benralizumab group was acute exacerbation twice a year and dupilumab group was acute exacerbation once a year, and this study used AER as the main end point. How to explain the reliability of the results? Although the three articles in this study are of good quality and have a large number cases enrolled, can there be a conclusion that dupilumab is superior to benralizumab in patients with higher blood eosinophil count, because there are only three papers and the inclusion criteria for the case were little different?Minor:
Line 103 Remove duplicate sentences: force expiratory volume at 1s Line 121(Figures 1 and 2) change to (Fugure 3A, 3B) Line 123-124 (RR and 95% CrI of 0.77 (0.52 to 1.15); Figure 3C). change to [RR and 95% CrI of 0.77 (0.52 to 1.15); Figure 3C]. Line 129 (Figure 2B, C) change to (Figure 3B, 3C) Line 132 (Figure 3) change to (Figure 3A, 3D) Line 175 2.4.1. Incidence of AAE change to 2.4.1. Incidence of AAE and SAE Line 179 1.023 (0.688–1.526) (Figure 6A). add : and there were no significant differences in the incidence of any SAEs between dupilumab or benralizumab and placebo (Figure 6B).Author Response
Response to Review report 2
Thank you for giving me the opportunity to submit a revised draft of my manuscript titled “Comparative efficacy and safety of dupilumab and benralizumab in patients with inadequately controlled asthma: a systematic review” to International journal of molecular sciences. We appreciate the time and effort that you have dedicated to providing your valuable feedback on my manuscript. We are grateful for your insightful comments on my paper. We have been able to incorporate changes to reflect most of the suggestions provided by the reviewers. We have highlighted (yellow) the changes within the manuscript.
Here is a point-by-point response to the reviewers’ comments and concerns.
Comments and Suggestions for Authors
This is a systemic literature review, it focused on comparing the efficacy and safety between dupilumab and benralizumab for inadequately controlled asthma. Initial 557 studies were enrolled and through selection process, only one dupilumab study and two benralizumab studies were identified after patients, interventions, comparison, outcome, and study designs (PICOS) were adopted. Using the Bayesian network meta-analysis statistical method with non-informative prior distribution, the results demonstrated that there was no significant difference in the annual exacerbation rate (AER) between dupilumab and benralizumab in overall patients and the subgroup with the blood eosinophil count of <150. However, the AER was significantly lower in the dupilumab group than in the benralizumab group in the subgroup with the blood eosinophil count of ≥150 but <300, and ≥300 with the rate ratio and 95% credible interval of 0.51 (0.29–0.92) and 0.58 (0.39–0.84), respectively. This indirect treatment comparison indicates that dupilumab is superior to benralizumab in patients with higher blood eosinophil count. This is the first paper to propose non-head-to-head comparing the efficacy and safety of dupilumab and benralizumab. Although the results were well described, however there are still some issues that must be clarified.
Comment 1
Major:
Although the authors mentioned that there were quite high-quality papers in this analysis, they also think that there was bias in the analysis of this study, but because their inclusion criteria were different in the severity degree, especially the benralizumab group was acute exacerbation twice a year and dupilumab group was acute exacerbation once a year, and this study used AER as the main end point. How to explain the reliability of the results? Although the three articles in this study are of good quality and have a large number cases enrolled, can there be a conclusion that dupilumab is superior to benralizumab in patients with higher blood eosinophil count, because there are only three papers and the inclusion criteria for the case were little different?
Response: We agree with this and have incorporated your suggestion. Thank you for highlighting this very important point. As rightly pointed out, the inclusion criteria differed between the dupilumab study and the benralizumab studies in terms of disease severity. In particular, the inclusion criteria with respect to the frequency of asthma exacerbations was at least once a year in the dupilumab group, but more than twice a year in the benralizumab group. In other words, the dupilumab group contained even milder cases than those in the benralizumab group. The dupilumab group showed better suppression of the frequency of exacerbations than benralizumab, even though it contained relatively milder cases. This heterogeneity among the included studies has been acknowledged as a study limitation in the discussion section. However, biologics for inadequately controlled asthma are generally more effective in severe cases and in patients with history of more frequent asthma exacerbations. Therefore, if the inclusion criteria for severity and frequency of past exacerbations were harmonized between the two groups, the difference between dupilumab and benralizumab with respect to suppression of asthma exacerbations would be greater. In other words, this heterogeneity does not seem to affect the conclusions made in this study. This has been added to the relevant part of the discussion section as follows:
Line290-295: “That is, the inclusion criteria with respect to disease severity differed between the two drugs. In particular, the benralizumab group had at least two exacerbations per year, while the dupilumab group had at least one exacerbation per year. Biologics have been reported to be particularly effective in patients with more severe disease and in those with more frequent exacerbations. If severity or frequency of exacerbations is standardized among the studies, the difference in efficacy between the two drug groups is expected to be more pronounced.”
Comment 2:
Minor:
Line 103 Remove duplicate sentences: force expiratory volume at 1s Line 121(Figures 1 and 2) change to (Figure 3A, 3B) Line 123-124 (RR and 95% CrI of 0.77 (0.52 to 1.15); Figure 3C). Change to [RR and 95% CrI of 0.77 (0.52 to 1.15); Figure 3C]. Line 129 (Figure 2B, C) change to (Figure 3B, 3C) Line 132 (Figure 3) change to (Figure 3A, 3D) Line 175 2.4.1. Incidence of AAE change to 2.4.1. Incidence of AAE and SAE Line 179 1.023 (0.688–1.526) (Figure 6A). Add: and there were no significant differences in the incidence of any SAEs between dupilumab or benralizumab and placebo (Figure 6B).
Response: Agree. Thank you for pointing out a very important issue. We have incorporated all the suggested changes in the revised manuscript as follows:
Line 103: The duplicate sentence “force expiratory volume at 1s” has been removed.
Line 122: “(Figures 1 and 2)” have been changed to “(Figure 3A, 3B)”
Line 124-125: “(RR and 95% CrI of 0.77 (0.52 to 1.15); Figure 3C)” has been changed to “[RR and 95% CrI of 0.77 (0.52 to 1.15); Figure 3C]”.
Line 130: “(Figure 2B, C)” has been changed to “(Figure 3B, 3C)”
Line 133: “(Figure 3)” has been changed to “(Figure 3A, 3D)”
Line 175: “Primary safety endpoint” has been changed to “Primary and secondary safety endpoint”
Line 176: “2.4.1. Incidence of AAE” has been changed to “2.4.1. Incidence of AAE and SAE”.
Line 180: Results of incidence of any severe adverse events have been added in this part. “2.5. Secondary safety endpoint” and the section of “2.5.1. Incidence of any severe adverse event (SAE)” has been removed because of duplication.
Line 192: “2.6. Bias assessment” has been changed to “2.5. Bias assessment”.
In addition to the above comments, all spelling and grammatical errors pointed out by the reviewers have been corrected.
We look forward to hearing from you in due time regarding our submission and to respond to any further questions and comments you may have.
Sincerely,
Koichi Ando
Department of Medicine, Division of Respiratory Medicine and Allergology, Showa University School of Medicine
1-5-8 Hatanodai, Shinagawa-ku, Tokyo, 142-8555, Japan
Tel: +81-3784-8532
Fax: +81-3-3784-8742
Email: koichi-a@med.showa-u.ac.jp